# Lack of Durable Remission with Conventional-Dose Total Skin Electron Therapy for the Management of Sezary Syndrome and Multiply Relapsed Mycosis Fungoides

**DOI:** 10.3390/cancers11111758

**Published:** 2019-11-08

**Authors:** Belinda A. Campbell, Gail Ryan, Christopher McCormack, Eleanor Tangas, Mathias Bressel, Robert Twigger, Odette Buelens, Carrie van der Weyden, H. Miles Prince

**Affiliations:** 1Department of Radiation Oncology, Peter MacCallum Cancer Centre, Melbourne 3000, Victoria, Australia; 2Department of Clinical Pathology, The University of Melbourne, Parkville 3010, Victoria, Australia; 3Department of Surgical Oncology, Peter MacCallum Cancer Centre, Melbourne 3000, Victoria, Australia; 4Department of Dermatology, St Vincent’s Hospital Melbourne, Fitzroy 3065, Victoria, Australia; 5Melbourne Clinical School, School of Medicine, University of Notre Dame, Werribee 3030, Victoria, Australia; 6Centre for Biostatics and Clinical Trials, Peter MacCallum Cancer Centre, Melbourne 3000, Victoria, Australia; 7Department of Clinical Haematology, Peter MacCallum Cancer Centre and Royal Melbourne Hospital, Melbourne 3000, Victoria, Australia; 8Sir Peter MacCallum Department of Oncology, The University of Melbourne, Parkville 3010, Victoria, Australia

**Keywords:** total skin electron therapy, cutaneous T-cell lymphoma, mycosis fungoides, Sezary syndrome, skin-directed therapy

## Abstract

Mycosis fungoides (MF) and Sezary syndrome (SS) are multi-relapsing, morbid, cutaneous T-cell lymphomas. Optimal treatment sequencing remains undefined. Total skin electron therapy (TSE) is a highly technical, skin-directed treatment, uniquely producing symptom-free and treatment-free intervals. Recent publications favour low-dose TSE for reduced toxicity, but early data support conventional-dose TSE (cdTSE) for longer disease control. Patient selection requires weighing-up tolerability against response durability. We investigated duration of benefit from cdTSE in patients with poorer prognosis diseases: SS and heavily pre-treated MF. Endpoints were overall survival, and “time to next treatment” (TTNT) as surrogate for clinical benefit duration. Seventy patients (53 MF, 17 SS) were eligible: median prior treatments, 4; median cdTSE dose, 30 Gy; median follow-up, 5.8 years. SS patients had worse prognosis (HR = 5.0, *p* < 0.001) and shorter TTNT (HR = 4.5, *p* < 0.001) than MF patients; median TTNT was only 3.7 months. Heavily pre-treated MF patients had inferior prognosis (HR = 1.19 per additional line, *p* = 0.005), and shorter TTNT (HR = 1.13 per additional line, *p* = 0.031). Median TTNT for MF patients with ≥3 prior treatments was 7.1 months, versus 23.2 months for 0–2 prior treatments. In conclusion, cdTSE has a limited role in SS. TTNT is reduced in heavily pre-treated MF patients, suggesting greater benefit when utilized earlier in treatment sequencing.

## 1. Introduction

Mycosis fungoides (MF) and Sezary Syndrome (SS) are diseases with long histories of multiple relapses and significant morbidity. The optimal management and sequencing of available treatments remain undefined [1,2,3,4]. Published guidelines highlight the complexity of treatment decision-making and the lack of standardised treatment algorithms for patients with MF/SS [5,6]. Total skin electron therapy (TSE) has been reported to be one of the most effective single agents for MF [7,8] although patient selection and endpoint definition undoubtedly play important roles in this assertion. TSE uniquely offers patients a symptom-free and highly desirable treatment-free interlude, however, the durability of benefit is of critical clinical importance, particularly when weighed against treatment toxicities and increasing availability of systemic treatment options. Prior to the development of skin-active systemic therapies, TSE represented the mainstay of MF treatment, with conventional-dose TSE (cdTSE, 30–36 Gy) achieving long-term disease control in patients with early stage disease in the first-line setting [9,10,11]. In current clinical practice, TSE is often deferred until later in the treatment paradigm, and frequently utilized to treat advanced, treatment-refractory disease, where the durability of disease control following TSE is uncertain.

Outside of a prospective clinical trial, establishing the exact date of disease relapse and/or progression is inherently difficult for patients with MF/SS. Time to next treatment (TTNT) represents a useful, measurable and clinically meaningful surrogate for the durability of clinical benefit, and features in the published literature as an accepted measure of effectiveness of systemic treatments for MF/SS [1,12,13,14,15]. In our previous report of an unselected cohort of MF/SS patients treated with cdTSE, the median TTNT was 7.8 months (95% CI, 4.4–14.7) [1]; the wide confidence interval suggesting varying durability of benefit across this group. Appropriate patient selection for cdTSE is critical for optimal duration of clinical benefit.

A recent shift in clinical practice has favoured low-dose TSE (10–12 Gy) over cdTSE for greater tolerability [16], accepting that the complete skin response rate and potential durability of response may be inferior based on early dose-effect data [9,11]. However, for patients seeking more durable symptom control and who are fit enough to tolerate cdTSE, the duration of clinical benefit becomes a pertinent question. To date, there is absence of data addressing this issue, and the role of cdTSE remains particularly controversial in poor prognostic patients with SS and heavily pre-treated MF [2]. This study investigates the durability of clinical benefit of cdTSE in these two patient groups with poor prognosis MF/SS.

## 2. Results

### 2.1. Characterisitics of Eligible Patients

A total of 136 “TSE-like” treatments were delivered in 122 patients during the study period, with 14 representing repeat treatments. Of the 122 patients receiving their first course of TSE, 43 were excluded for “partial body” and/or low-dose TSE, with a further 9 excluded for insufficient treatment details. Seventy patients were eligible: 52 (74.3%) had biopsy-confirmed MF, 1 (1.4%) had PTCL-NOS, and 17 (24.3%) had SS. (Table 1) Seven (10.0%) patients had high-grade transformation prior to cdTSE. At time of cdTSE, median age was 66 (range, 28.6–83.9) years. The extent of skin disease was T1-2 in 35 (50.0%) patients, and T3-4 in 35 (50.0%); 57 (90.4%) were ECOG 0-1 (Table 1). Patients received a variety of treatments prior to cdTSE, with median of 4 (range, 0–14) failed treatment lines (excluding topical steroids) (Table 2). cdTSE was delivered first-line in 7 (10.0%) patients. Five patients (3 high-grade transformed MF, 2 SS) received cdTSE as a planned adjuvant to stem cell transplantation (SCT): high-dose chemotherapy with autologous SCT (3); total body irradiation with autologous SCT (1); bortezomib and allogeneic SCT (1). Four of the 5 patients were planned to receive high dose cytotoxic therapy with autologous haematopoietic SCT followed by cdTSE, however, all patients had residual, active skin disease at the time of cdTSE. One patient was planned to receive cdTSE prior to conditioning chemotherapy and allogeneic haematopoietic SCT, however, this patient had extremely short-lived duration of disease response to cdTSE, with frank progression of skin disease occurring between cdTSE and commencement of chemotherapy and allogeneic SCT.

### 2.2. Conventional-Dose Total Skin Electron Therapy

All patients were initially planned to receive cdTSE (30–36 Gy). Median fraction size was 1.5 Gy (range, 1.2–4 Gy). The two most common fractionation schedules were 3 fractions per week (48.6%), and 5 fractions per week with a planned 2-week break after 15 Gy (47.1%). Shielding was used in 48 patients to minimise toxicity in sensitive areas with no prior lymphomatous involvement: 31 (44.3%) had head/scalp shielding, 17 (24.3%) had nail bed and/or eyelid shielding; and 2 (2.9%) also had feet shielding. Five patients failed to complete cdTSE (receiving 12–24 Gy): two due to severe infections (multilobar pneumonia, 1, and endocarditis, 1), one due to atrial fibrillation and acute cardiac failure, and two unknown. One patient declined to complete the prescribed boosts due to acute toxicities of cdTSE (fatigue and skin discomfort). Two additional patients required unscheduled breaks in cdTSE delivery: one due to myocardial ischaemia and acute cardiac failure, and one due to sepsis secondary to lost skin integrity with concurrent influenza infection.

### 2.3. Overall Survival of the Whole Cohort

Median follow-up was 5.8 years. For the whole cohort, the OS was 83% (95% CI: 71–90) at 12 months, and 76% (95% CI: 64–85) at 24 months (Figure 1a). The percentages of patients not requiring further therapy at 12 months and 24 months were 39% (95% CI: 26–51) and 22% (95% CI: 13–34), respectively (Figure 1b). This analysis was repeated, excluding patients receiving SCT (*n* = 5), with similar results: the treatment-free rate was 40% (95% CI: 27–53%) at 12 months, and 24% (95% CI: 14–37) at 24 months.

### 2.4. Overall Survival and Time to Next Treatment in Patients with Sezary Syndrome

Patient characteristics for SS (*n* = 17) and MF patients (*n* = 53) are listed in Table 1. The median number of prior therapies was 7 (range, 0–13) for SS, and 3 (range, 0–14) for MF patients. Following cdTSE, patients with SS had inferior OS compared to MF patients (HR = 5.0, 95% CI: 2.4–10.2; *p* < 0.001). Median OS for SS and MF patients were 18.8 months (95% CI: 6.2–30.9) versus 57.9 (95% CI: 40.2–107); and 5–year OS were 0% versus 50% (95% CI: 33–64), respectively (Figure 2a).

Initially, patients with SS and MF derived similar skin benefits from cdTSE: overall skin response rates were 93.8% versus 92.1%, respectively (Table 1). However, durability of clinical benefit was notably inferior for SS patients compared to MF patients (HR = 4.5, 95% CI: 2.2–9.2, *p* < 0.001). Following cdTSE, median TTNT was 3.7 months (95% CI: 2.3–4.4) versus 10.9 months (95% CI: 5.1–20.3) for patients with SS and MF, respectively (Figure 2b). All SS patients required further therapy within 9 months of cdTSE. The poor TTNT observed in this group of SS patients appeared to be unrelated to the number of prior treatments (Figure 2c).

### 2.5. Overall Survival and Time to Next Treatment in Heavily Pre-Treated Patients with Mycosis Fungoides, According to Number of Prior Lines of Therapy

In the 53 MF patients, prognosis deteriorated with increasing exposure to previous therapies. Adjusting by time from diagnosis until commencement of cdTSE, the HR per additional prior treatment line was 1.19 (95% CI: 1.06–1.35, *p* = 0.005). After cdTSE, median OS for patients with 0–2 versus ≥3 prior treatment lines were 80.6 months (95% CI: 37.6- not estimable) versus 45.8 months (95% CI: 30.1–78.1), with 5-year OS of 68% (95% CI: 42–84) and 34% (95% CI: 15–54), respectively (Figure 3a).

Following cdTSE, heavily pre-treated MF patients had inferior duration of clinical benefit. For TTNT, the HR per additional prior treatment line was 1.13 (95% CI: 1.01–1.27, *p* = 0.031), when adjusted by time from diagnosis until commencement of cdTSE. Median TTNT was 23.2 (95% CI: 12.7–34.8) for patients with 0–2 prior lines of therapy, versus 7.1 (95% CI: 3.4–10.9) months for patients with ≥3 previous treatments. At 12 months, the proportions of patients not requiring further therapy were 76% (95% CI: 51–89) and 31% (95% CI: 15–49), for patients with 0–2 prior treatments and ≥3 prior treatments, respectively (Figure 3b).

## 3. Discussion

TSE is a highly technical, skin-directed therapy, with proven efficacy in the treatment of MF. Following cdTSE, 50% of MF patients in this study benefitted from a treatment-free interval greater than one year. Comparing the results of this study with published data [1] confirms the clinical value of cdTSE in the modern therapeutic armamentarium: a previous analysis from our group revealed a median TTNT from all systemic treatments of only 5.4 months (95% CI: 5.1–6.1) [1] (Table 3). However, the increasing availability of skin-active therapies for MF/SS has meant that TSE is now frequently deferred until later in the treatment paradigm, and often utilised for multiply relapsed or treatment-refractory disease [2]. Geographic availability of TSE and referral patterns may also impact on utilisation of this treatment. Presently, TSE dose, patient selection and treatment sequencing remain contentious clinical dilemmas.

Historically, durability of disease response following TSE was found to be dose-dependent, with cdTSE associated with longer disease-free survival for MF patients [9]. However, in modern clinical practice, low-dose TSE is increasingly favoured for reduced toxicity risks, shorter course, and ease of re-treatment. To date, no randomised trials have evaluated the durability of disease control achieved by low-dose TSE versus cdTSE. In the modern literature, four single-arm studies of low-dose TSE (10–12 Gy) have reported median response/clinical benefit durations of 5.2–16.4 months in patients with MF/SS [2,17,18,19,20]. A Phase II study of very low-dose TSE (4 Gy) in stage IB-II MF patients revealed a disappointing median time to relapse of 2.7 months [21]. Smaller, retrospective series (including 11–26 patients with MF) have attempted to investigate the TSE dose-effect on event-free survival and/or duration of clinical benefit, with conflicting conclusions [8,22,23,24,25,26]. Notably, these studies did not account for the effect of prior treatment lines on the durability of benefit, and some studies also admixed SS patients into the analyses, potentially confounding the results. Translating historical data into modern clinical practice is difficult but, drawing from earlier dose-response data [9,10,11], it is possible that low-dose TSE is associated with inferior durability of clinical benefit and shorter treatment-free intervals than cdTSE. Larger, prospective studies are required to fully investigate the impact of TSE dose on durability of benefit and quality of life endpoints in the current treatment era of heavily pre-treated patients with MF/SS.

This study explored the durability of palliative benefit from cdTSE in SS patients. SS is an aggressive disease entity, with significant morbidity and poor prognosis. cdTSE provided SS patients early relief from their cutaneous disease, with both MF and SS patients achieving overall skin response rates >92%. However, the duration of clinical benefit was significantly inferior in SS patients (HR = 4.5, *p* < 0.001), with median TTNT of only 3.7 months. SS patients tended to be heavily pre-treated, with a median of 7 failed therapies prior to cdTSE, but the poor TTNT in this group appears to be unrelated to prior treatment exposures, as even SS patients receiving cdTSE early in the treatment paradigm experienced short clinical benefit. Given that the course of cdTSE takes approximately 2 months to deliver and is associated with clinically significant toxicity risks, this short-lived benefit is extremely disappointing. It is possible that the short TTNT may be related to the untreated systemic disease during the course of skin-directed cdTSE, and therefore newer systemic therapies with improved therapeutic ratio may offer more hope for patients with SS. A recently published study from our group found extracorporeal photophoresis (ECP) monotherapy to be an effective and well-tolerated systemic treatment for SS, with patients achieving an impressive median TTNT of 14 months, significantly exceeding that of other available systemic therapies, including interferon, HDAC inhibitors, chemotherapy, low-dose methotrexate and other immunomodulatory agents [15]. The monoclonal antibody mogamulizumab also has impressive efficacy against SS, with median TTNT of 12.9 months reported from the phase 3 MAVORIC study (NCT01728805) [13]. We believe that well-tolerated, effective systemic therapies should be considered in preference to cdTSE for patients with SS.

This study is the first to report the inferior prognosis of multiply relapsed and heavily pre-treated MF patients treated with cdTSE. Following cdTSE, 5-year OS was halved in patients with prior exposure to ≥3 treatment lines: 34% versus 68% for patients with 0–2 prior lines. Furthermore, this is the first study to demonstrate that increasing exposure to prior treatment is a negative predictor for the durability of clinical benefit in MF patients receiving cdTSE, with 13% decrement in TTNT per line of previous therapy (*p* = 0.031). Despite this inferior result, there was still benefit from cdTSE in MF patients with ≥3 previous treatment lines, with TTNT in keeping with the reported durability of disease control of other available therapies [1] (Table 3). However, the diminishing benefit in heavily pre-treated MF patients becomes an important consideration in patient selection for cdTSE. Further research is required to investigate whether this negative effect on TTNT also applies to recipients of low-dose TSE.

With widening availability of novel therapies, the optimal treatment sequencing for MF/SS becomes an increasingly important question. For MF patients, we found that earlier delivery of cdTSE (0–2 prior treatment lines) produced an excellent median TTNT of 23.2 months, with 76% of patients remaining treatment-free at 1 year, representing an impressive treatment-free interval. However, MF patients receiving cdTSE later in the treatment paradigm achieved shorter clinical benefit. This phenomenon has also been observed in a cohort of 65 SS patients receiving monotherapy ECP: patients receiving early ECP (in lines 1–3) experienced longer median TTNT than those receiving later ECP (>3 lines): 12 months versus 7 months, (*p* = 0.07) [15]. Interestingly, this effect of treatment timing on TTNT was not observed in MF/SS patients receiving other systemic therapies: a retrospective study of 88 MF/SS patients treated with a range of systemic therapies found no statistically significant difference in the TTNT of early delivery (lines 1–2) versus later delivery (>2 lines) for interferon, HDAC inhibitors, and chemotherapy [12]. The mechanism for this observed reduction in cdTSE efficacy is unclear; one hypothesis is that increasing exposure to prior therapies alters the biology of the disease producing a more aggressive and radio-resistant phenotype. These findings merit further investigation but suggest that cdTSE should be delivered earlier in the treatment paradigm to optimise the treatment-free interval for patients with MF. Given the increasing use of low-dose TSE in current clinical practice, we look forward to extending our analyses to more recently treated MF patients receiving low-dose TSE.

We acknowledge the limitations of this retrospective study with restricted cohort size. Strengths of this study include central pathology review, careful review of irradiated skin volumes, and consistency in cdTSE technique over the study period. With uniformity across the cohort relating to patterns of practice and decision-making by a super-specialised and stable multidisciplinary team, we are confident that TTNT represents a reliable surrogate for the duration of clinical benefit. Future investigations into optimal treatment sequencing will benefit from prospective study design and larger population. Further, the new, collaborative PROCLIPI (PROspective Cutaneous Lymphoma International Prognostic Index) registry database will allow for larger cohort studies, with the ultimate aim to create clinical-based algorithms to guide management decisions [27].

## 4. Materials and Methods

The Peter MacCallum Cancer Centre Ethics Committee approved this study. Eligibility required pathologically confirmed diagnoses of MF/SS, and cdTSE (30–36 Gy) prescribed between 1 June 1998 and 1 September 2016. All diagnoses were confirmed by central pathology review. Exclusion criteria included cutaneous B-cell and T-cell non-Hodgkin lymphomas with secondary cutaneous involvement. Charts of eligible patients were retrospectively reviewed for information regarding demographics, clinical features, prognostic factors, treatment details and outcomes. Staging [28] was determined at time of diagnosis, with “re-staging” prior to delivery of cdTSE. (Table 4) For patients receiving repeated TSE, only the first course of TSE was considered in this study. The analyses were performed as “intention to treat”. No patients originally planned to receive low-dose (defined as <20 Gy) or “partial body” TSE (defined as shielding of >10% body surface area) were included in the analyses. For quality assurance, all patient charts were carefully reviewed to confirm the prescribed TSE dose and irradiated skin volume.

All cdTSE were delivered using a rotary-dual technique, the patient standing on a rotating platform, alternating arm and leg positions, with maximal skin exposure. In-vivo thermoluminescent dosimeters were used to assess cutaneous dose, particularly in sites likely to be under-dosed due to self-shielding. Standard fractionation schedules were (i) 3 fractions per week, or (ii) 5 fractions per week with a planned 2-week break after 15 Gy; selection of fractionation schedule considered patient convenience and preference. cdTSE dose was prescribed to the skin surface, with 90% expected at 5mm. All attempts were made to maintain dose heterogeneity within +/−10% of the prescribed dose. Tumours too deep for adequate treatment with cdTSE were first debulked using local electron or superficial orthovoltage fields. Self-shielded areas of skin received supplementary “boosts” using localised electron fields; boosts were usually hypo-fractionated (3–4 Gy per fraction), with prescribed dose individualised to in-vivo dosimetry measurements.

Cutaneous response to treatment was determined at follow-up visits 3–8 weeks after completion of cdTSE and boosts. Overall skin response was defined as complete or partial clearance of clinically visible cutaneous disease. Stable disease was defined as no apparent change in cutaneous involvement. Disease progression was defined as an increase in cutaneous disease. The more recently defined response criteria could not be retrospectively applied in this study cohort [29].

In this study, TTNT was used as the surrogate endpoint for time to treatment failure after cdTSE, and as a marker of the duration of clinical benefit. TTNT was measured from the date of commencing cdTSE to the date of the next treatment course. For purposes of this study, topical corticosteroids were not considered as separate treatment courses. For patients not receiving a subsequent treatment, TTNT was censored at the date of commencement of palliative care, death or last follow-up. Overall survival (OS) was measured from the date of commencing cdTSE until the date of death from any cause; living patients were censored at date of last follow-up. The Kaplan-Meier method was used to estimate TTNT and OS. Cox proportional hazards regression was used to model the association between time-to-event outcomes and number of previous treatment lines, with p-values based on the likelihood ratio test. Analyses were adjusted by time from diagnosis until commencement of cdTSE. Patients were dichotomised into 0–2 versus ≥3 prior lines of therapy, for descriptive purposes. Given the limited numbers of patients and events, multivariable analyses were not performed. All statistical analyses were performed in R 3.4.2. [30].

## 5. Conclusions

With appropriate patient selection, skin-directed TSE uniquely offers MF/SS patients symptom-free and treatment-free intervals. However, we have shown that the durability of clinical benefit from cdTSE is inferior in patients with poor prognosis disease. For patients with SS, cdTSE is of limited value with short TTNT and poor survival. For MF patients receiving cdTSE, this is the first study to demonstrate the negative effect of increased exposure to prior therapies. Multiply relapsed and heavily pre-treated MF patients represent a poorer prognostic group with diminishing TTNT following cdTSE. Earlier delivery of cdTSE in the treatment paradigm may benefit patients with MF. Future investigation is required to extend this analysis to low-dose TSE, determine the optimal sequencing of available therapies, and explore new systemic therapies with lasting therapeutic value and low toxicity profiles for patients with poor prognosis MF/SS.

## Figures and Tables

**Figure 1 cancers-11-01758-f001:**
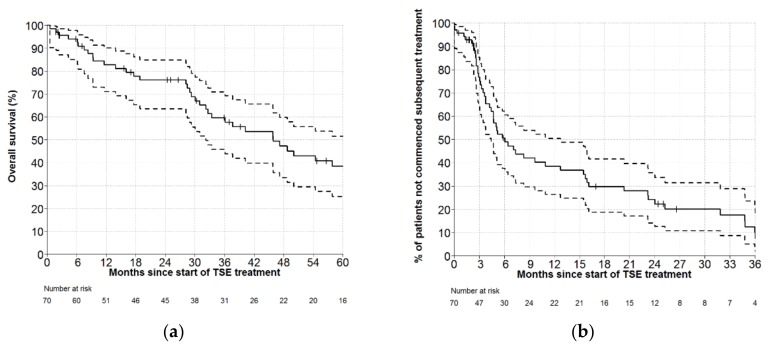
(**a**) Overall survival following conventional-dose total skin electron therapy (TSE) for the whole cohort. Dashed lines represent the 95% confidence interval. (**b**) Time to next treatment following conventional-dose total skin electron therapy (TSE) for the whole cohort. Dashed lines represent the 95% confidence interval.

**Figure 2 cancers-11-01758-f002:**
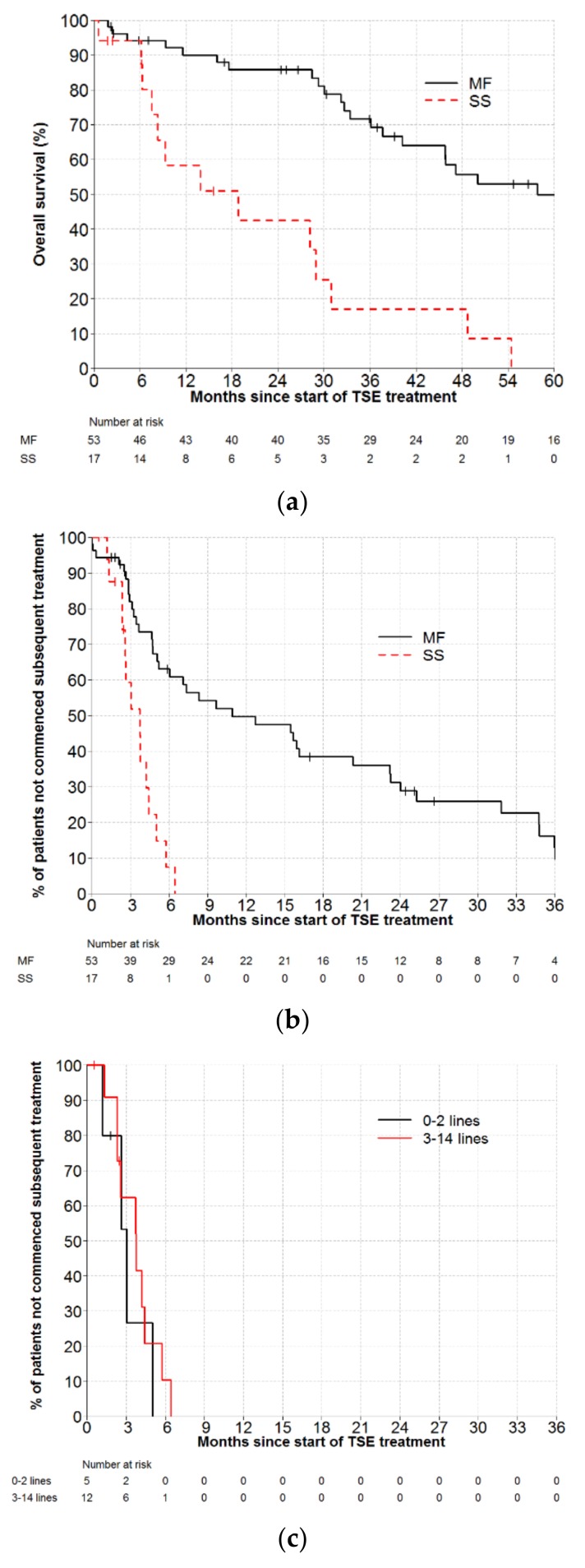
(**a**) Overall survival for patients with mycosis fungoides (MF) and Sezary syndrome (SS) treated with conventional-dose total skin electron therapy (TSE), HR = 5.0 (95% CI: 2.4–10.2, *p* < 0.001). (**b**) Time to next treatment for patients with mycosis fungoides (MF) and Sezary syndrome (SS) treated with conventional-dose total skin electron therapy (TSE), HR = 4.5 (95% CI: 2.2–9.2, *p* < 0.001). (**c**) Time to next treatment by number of previous lines of therapy in patients with Sezary syndrome treated with conventional-dose total skin electron therapy (TSE).

**Figure 3 cancers-11-01758-f003:**
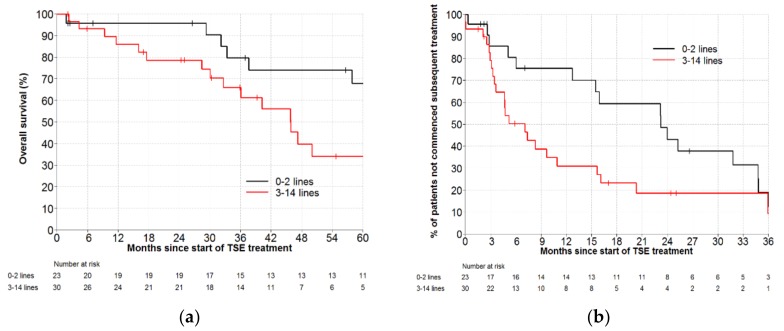
(**a**) Overall survival by number of previous lines of therapy in patients treated with conventional-dose total skin electron therapy (TSE) for mycosis fungoides, HR per additional line of treatment = 1.19 (95% CI: 1.06–1.35, *p* = 0.005). (**b**) Time to next treatment by number of previous lines of therapy in patients treated with conventional-dose total skin electron therapy (TSE) for mycosis fungoides, HR per additional previous line of treatment = 1.13 (95% CI: 1.01–1.27, *p* = 0.031).

**Table 1 cancers-11-01758-t001:** Patient, tumour and treatment characteristics.

Variable	Total (*n* = 70)	MF (*n* = 53)	SS (*n* = 17)
**Gender**			
Female	28 (40.0%)	19 (35.8%)	9 (52.9%)
Male	42 (60.0%)	34 (64.2%)	8 (47.1%)
**Age at diagnosis (years)**			
Median (range)	66 (14–81)	63 (14–81)	68 (49–81)
**Diagnosis**			
Mycosis fungoides (MF)	52 (74.3%)	52 (98.1%)	0 (0.0%)
Peripheral T-cell lymphoma, not otherwise specified (PTCL-NOS)	1 (1.4%)	1 (1.9%)	0 (0.0%)
Sezary syndrome (SS)	17 (24.3%)	0 (0.0%)	17 (100.0%)
**Subtype**			
Folliculotropic	7 (10.0%)	7 (13.2%)	0 (0.0%)
Poikilodermatous	1 (1.4%)	1 (1.9%)	0 (0.0%)
Subtype not specified	62 (88.6%)	45 (84.9%)	17 (100.0%)
**Large cell transformation**			
Prior to cdTSE	7 (10.0%)	6 (11.3%)	1 (5.9%)
After cdTSE	11 (15.7%)	9 (17.0%)	2 (11.8%)
**Stage at diagnosis (missing *n* = 1) ^1^**			
IA	5 (7.1%)	5 (9.4%)	0 (0.0%)
IB	24 (34.3%)	23 (43.4%)	1 (5.9%)
IIA	2 (2.9%)	2 (3.8%)	0 (0.0%)
IIB	17 (24.3%)	16 (30.2%)	1 (5.9%)
III	10 (14.3%)	4 (7.5%)	6 (35.3%)
IVA1	8 (11.4%)	1 (1.9%)	7 (41.2%)
IVA2	2 (2.9%)	1 (1.9%)	1 (5.9%)
IVB	2 (2.9%)	1 (1.9%)	1 (5.9%)
**ECOG at diagnosis (missing *n* = 3)**			
0	57 (85.1%)	47 (92.2%)	10 (62.5%)
1	10 (14.9%)	4 (7.8%)	6 (37.5%)
**T Stage at cdTSE**			
1	1 (1.4%)	1 (1.9%)	0 (0.0%)
2	34 (48.6%)	32 (60.4%)	2 (11.8%) ^2^
3	15 (21.4%)	14 (26.4%)	1 (5.9%) ^2^
4	20 (28.6%)	6 (11.3%)	14 (82.4%)
**ECOG at cdTSE (missing *n* = 7)**			
0	36 (57.1%)	33 (71.7%)	3 (17.6%)
1	21 (33.3%)	10 (21.7%)	11 (64.7%)
2	4 (6.3%)	2 (4.3%)	2 (11.8%)
3	2 (3.2%)	1 (2.2%)	1 (5.9%)
**Treatment lines prior to cdTSE**			
Median (range)	4 (0–14)	3 (0–14)	7 (0–13)
**Fractionation schedule**			
5 fractions per week	33 (47.1%)	22 (41.5%)	11 (64.7%)
4 fractions per week	1 (1.4%)	1 (1.9%)	0 (0.0%)
3 fractions per week	34 (48.6%)	28 (52.8%)	6 (35.3%)
2 fractions per week	2 (2.9%)	2 (3.8%)	0 (0.0%)
**Unscheduled treatment disruptions or failure to complete cdTSE**			
Yes ^3^	8 (11.4%)	5 (9.4%)	3 (17.6%)
**Skin response following cdTSE (missing *n* = 3)**			
Overall skin response (complete or partial)	62 (92.5%)	48 (92.2%)	14 (93.8%)
Stable disease	4 (6.0%)	3 (5.9%)	1 (6.3%)
Progression	1 (1.5%)	1 (2.0%)	0 (0.0%)

cdTSE = conventional-dose total skin electron therapy; ECOG = Eastern Cooperative Oncology Group performance state; ^1^ For patients with secondary SS, “stage at diagnosis” reflects the original stage of disease at the time of diagnosis with MF; ^2^ At time of cdTSE, 3 previously treated SS patients no longer had T4 generalised erythroderma; ^3^ Reasons were: acute myocardial ischaemia and cardiac failure, 1; sepsis secondary to loss of skin integrity plus concurrent influenza infection, 1; fatigue and painful skin erythema requiring oral steroid therapy, 1; multi-lobar pneumonia, DVT and cellulitis of the foot, 1; acute endocarditis requiring urgent aortic valve surgery, 1; atrial fibrillation and cardiac failure, 1; unknown, 2.

**Table 2 cancers-11-01758-t002:** Treatments received prior to conventional-dose total skin electron therapy (cdTSE), excluding topical steroids.

Therapy	Total (*n* = 70)	MF (*n* = 53)	SS (*n* = 17)
Phototherapy ^1^	43 (61.4%)	34 (64.2%)	9 (52.9%)
Methotrexate	34 (48.6%)	21 (39.6%)	13 (76.5%)
Systemic corticosteroids	32 (45.7%)	20 (37.7%)	12 (70.6%)
Interferon	23 (32.9%)	14 (26.4%)	9 (52.9%)
Localised radiotherapy	20 (28.6%)	19 (35.8%)	1 (5.9%)
Multi-agent chemotherapy	16 (22.9%)	8 (15.1%)	8 (47.1%)
Histone deacetylase (HDAC) inhibitor	13 (18.6%)	7 (13.2%)	6 (35.3%)
Single-agent chemotherapy	11 (15.7%)	7 (13.2%)	4 (23.5%)
Extracorporeal photopheresis	9 (12.9%)	0 (0%)	9 (52.9%)
Haematopoietic stem cell transplant ^2^	5 (7.1%)^4^	3 (5.7%)	2 (11.8%)
Immunosuppressant ^3^	5 (7.1%)	4 (7.5%)	1 (5.9%)
Monoclonal antibodies ^4^	5 (7.1%)	3 (5.7%)	2 (11.8%)
Retinoid	5 (7.1%)	3 (5.7%)	2 (11.8%)
Denileukin difitox	2 (2.9%)	0 (0%)	2 (11.8%)
Bromodomain and extra-terminal motif (BET) inhibitor	1 (1.4%)	1 (1.9%)	0 (0%)
Bortezomib	1 (1.4%)	1 (1.9%)	0 (0%)
No prior therapy (cdTSE first-line)	7 (10.0%)	6 (11.3%)	1 (5.9%)

MF = mycosis fungoides; SS = Sezary syndrome; ^1^ PUVA, UVA, UVB; ^2^ Includes 4 patients who received pre-planned treatment packages consisting of autologous haematopoietic SCT followed by cdTSE (however, all patients had residual, active disease present at the time of cdTSE), and 1 patient who had earlier treatment with autologous haematopoietic SCT occurring 29.5 months prior to cdTSE; ^3^ azothiaprine, leflunomide; ^4^ alemtuzumab, infliximab.

**Table 3 cancers-11-01758-t003:** Comparison of the time to next treatment (TTNT) following conventional-dose total skin electron therapy (cdTSE) in this cohort and the TTNT of other therapies as reported in a previous publication from our group [1].

Treatment	n	Median TTNT (Months)	TTNT 95% CI (Months)	1-Year Free from Further Treatment (%)	2-Years Free from Further Treatment (%)	Median Number of Previous Lines of Therapy
a-interferon	68	8.7	6.0–18.0	41.7	29.1	1
Low-dose methotrexate	83	5.0	3.6–6.5	25.1	21.2	1
Histone deacetylase (HDAC) inhibitors	74	4.5	4.0–6.1	20.0	14.5	2
Bexarotene	20	7.3	2.6–110.8	47.4	36.8	1
Alemtuzumab	16	4.1	2.7–6.5	27.8	27.8	2.5
Denileukin diftitox	22	5.1	2.7–6.5	22.7	22.7	3
Chemotherapy	143	3.9	3.2–5.1	10.7	5.4	3
Extracorporeal photophoresis (ECP)	53	9.2	5.9–12.8	39.1	25.7	1
Autologous SCT	19	7.8	4.7–24.4	41.5	28.4	2
Allogeneic SCT	9	34.6	11.5–NA	80.0	53.3	5
cdTSE for MF ^1^ (current study)	23	23.2	12.7–34.8	76	43	0–2
	30	7.1	3.4–10.9	31	19	3–14
cdTSE for SS (current study)	17	3.7	2.3–4.4	0	0	7

SCT = haematopoietic stem cell transplantation; NA = not able to be assessed; MF = mycosis fungoides; SS = Sezary syndrome. ^1^ according to number of lines of previous therapy.

**Table 4 cancers-11-01758-t004:** ISCL/EORTC revision to the classification of skin involvement in patients with mycosis fungoides and Sezary syndrome [28].

Skin Stage	Description
T_1_	Limited patches, papules, and/or plaques covering <10% of the skin surface. May further stratify into T_1a_ (patch only) vs. T_1b_ (plaque ± patch).
T_2_	Patches, papules or plaques covering ≥10% of the skin surface. May further stratify into T_2a_ (patch only) vs. T_2b_ (plaque ± patch).
T_3_	One or more tumours (≥1 cm diameter)
T_4_	Confluence of erythema covering ≥80% body surface area

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
