# Peer review of "Lack of Durable Remission with Conventional-Dose Total Skin Electron Therapy for the Management of Sezary Syndrome and Multiply Relapsed Mycosis Fungoides"

_cancers, 2019, doi:10.3390/cancers11111758_

Round 1

Reviewer 1 Report

This is an interesting article which may help clinicians involved in the treatment of cutaneous T-cell lymphomas. In particular, it indicates that conventional-dose total skin electron therapy is not generally appropriate for advanced mycosis fungoides. The study is well structured and conclusions are consistent with the clinical data. 

Author Response

We would like to thank Reviewer 1 for their appraisal of our manuscript and for their positive feedback.

Reviewer 2 Report

This is a well written paper who put in place the usefulness of cd Total skin electron therapy in patients with MF and the limited use of it in Sezary syndrome. 

Specific comments 

Page 2 Characteristic of eligible patients line 84,85 please clarify the difference between adjuvant stem cells therapy and autologous BMT in the cohort.

Table 3 is not clear 

If based on current cohort please specify the nu of patients in each first line therapy.

If based on literature please provide ref and number  of pts in the cohort.

Author Response

We thank this Reviewer 2 for their positive and constructive feedback.  We have adopted these excellent suggestions, as follows:

(1) Page 2 Characteristic of eligible patients line 84,85 please clarify the difference between adjuvant stem cells therapy and autologous BMT in the cohort.

Thank you for this astute comment. 

(a) In our cohort, 5 patients received cdTSE as part of a pre-planned treatment package with haematopoietic stem cell transplantation (SCT).  However, all 5 patients had active skin disease at the time of TSE.  Four of the 5 patients were planned to receive high dose cytotoxic therapy with autologous haematopoietic SCT followed by cdTSE, however, all patients had residual, active skin disease at the time of cdTSE.  One patient was planned to receive cdTSE prior to conditioning chemotherapy and allogeneic haematopoietic SCT, however, this patient had extremely short-lived duration of disease response to cdTSE, with frank progression of skin disease occurring between cdTSE and commencement of chemotherapy and allogeneic SCT. In the interests of full disclosure, we have described these 5 patients as receiving cdTSE as a planned adjuvant to SCT.  We have added the following to the text:

Four of the 5 patients were planned to receive high dose cytotoxic therapy with autologous haematopoietic SCT followed by cdTSE, however, all patients had residual, active skin disease at the time of cdTSE.  One patient was planned to receive cdTSE prior to conditioning chemotherapy and allogeneic haematopoietic SCT, however, this patient had extremely short-lived duration of disease response to cdTSE, with frank progression of skin disease occurring between cdTSE and commencement of chemotherapy and allogeneic SCT.”

(b) Table 2 describes the therapies received prior to cdTSE in this cohort of patients.  As listed in Table 2, 5 patients received haematopoietic SCT prior to cdTSE.  These 5 patients include the afore mentioned 4 patients who received autologous SCT prior to cdTSE as part of a pre-planned treatment package.  One additional patient had previous treatment with autologous SCT occurring 29.5 months prior to cdTSE.  We have added the following note to Table 2:

“Includes 4 patients who received pre-planned treatment packages consisting of autologous SCT followed by cdTSE (however, all patients had residual, active disease present at the time of cdTSE), and 1 patient who had earlier treatment with autologous SCT occurring 29.5 months prior to cdTSE.

(2) Table 3 is not clear 

If based on current cohort please specify the nu of patients in each first line therapy.

If based on literature please provide ref and number of pts in the cohort.

Thank you for this excellent suggestion to improve Table 3. 

(a) We have clarified the caption for Table 3:

"Table 3. Comparison of the time to next treatment (TTNT) following conventional-dose total skin electron therapy (cdTSE) in this cohort, and the TTNT of other therapies as reported in a previous publication from our group[1]"

(b) We have also added the numbers of patients from the current study and published literature.  Please see the attached Word document, with changes highlighted in yellow.

Reviewer 3 Report

This is a well-written manuscript reporting the efficacy of conventional dose total skin electron beam (TSEB) radiation on multiply relapsed mycosis fungoides (MF)and Sezary syndrome (SS) patients. Prior studies reporting the benefit of TSEB focused on percentage of remission and duration of remission achieved. This paper included analysis of time to next treatment which perhaps is becoming a more meaningful endpoint in cutaneous lymphoma patients. Although the cohort of SS patients was not very large, the study was able to demonstrate poor durability of response in this group. The study design was well thought out and appropriate conclusions were discussed.

Author Response

We would like to thank Reviewer 3 for their appraisal of our manuscript and for their positive feedback.